# Evaluating the population impact of hepatitis C direct acting antiviral treatment as prevention for people who inject drugs (EPIToPe) – a natural experiment (protocol)

Matthew Hickman,[1] John F Dillon,[2] Lawrie Elliott,[3] Daniela De Angelis,[4] Peter Vickerman,[1] Graham Foster,[5,6] Peter Donnan,[7] Ann Eriksen,[8] Paul Flowers,[3] David Goldberg,[3,9] William Hollingworth,[1] Samreen Ijaz,[10] David Liddell,[11] Sema Mandal,[10] Natasha Martin,[12] Lewis J Z Beer,[13] Kate Drysdale,[5,6] Hannah Fraser,[1] Rachel Glass,[10] Lesley Graham,[14] Rory N Gunson,[15] Emma Hamilton,[16] Helen Harris,[10] Magdalena Harris,[17] Ross Harris,[10] Ellen Heinsbroek,[10] Vivian Hope,[18] Jeremy Horwood,[1] Sarah Karen Inglis,[13] Hamish Innes,[3,9] Athene Lane,[1] Jade Meadows,[1] Andrew McAuley,[3,9] Chris Metcalfe,[1] Stephanie Migchelsen,[10] Alex Murray,[16] Gareth Myring,[1] Norah E Palmateer,[3,9] Anne Presanis,[4] Andrew Radley,[2,19] Mary Ramsay,[10] Pantelis Samartsidis,[4] Ruth Simmons,[10] Katy Sinka,[10] Gabriele Vojt,[3] Zoe Ward,[1] David Whiteley,[20] Alan Yeung,[3,9] Sharon J Hutchinson[3,9]

For numbered affiliations see end of article.

**Correspondence to**
Prof Matthew Hickman;
matthew.hickman@bristol.ac.uk

## ABSTRACT

**Introduction** Hepatitis C virus (HCV) is the second largest contributor to liver disease in the UK, with injecting drug use as the main risk factor among the estimated 200 000 people currently infected. Despite effective prevention interventions, chronic HCV prevalence remains around 40% among people who inject drugs (PWID). New direct-acting antiviral (DAA) HCV therapies combine high cure rates (>90%) and short treatment duration (8 to 12 weeks). Theoretical mathematical modelling evidence suggests HCV treatment scale-up can prevent transmission and substantially reduce HCV prevalence/ incidence among PWID. Our primary aim is to generate empirical evidence on the effectiveness of HCV 'Treatment as Prevention' (TasP) in PWID.

**Methods and analysis** We plan to establish a natural experiment with Tayside, Scotland, as a single intervention site where HCV care pathways are being expanded (including specialist drug treatment clinics, needle and syringe programmes (NSPs), pharmacies and prison) and HCV treatment for PWID is being rapidly scaled-up. Other sites in Scotland and England will act as potential controls. Over 2 years from 2017/2018, at least 500 PWID will be treated in Tayside, which simulation studies project will reduce chronic HCV prevalence among PWID by 62% (from 26% to 10%) and HCV incidence will fall by approximately 2/3 (from 4.2 per 100 person-years (p100py) to 1.4 p100py). Treatment response and re-infection rates will be monitored. We will conduct focus groups and interviews with service providers and patients that accept and decline treatment to identify barriers and facilitators in implementing TasP. We will conduct longitudinal interviews with up to 40 PWID to assess whether successful HCV treatment alters their perspectives on and engagement with drug treatment and recovery. Trained peer researchers will be involved in data collection and dissemination. The primary outcome – chronic HCV prevalence in PWID – is measured using information from the Needle Exchange Surveillance Initiative survey in Scotland and the Unlinked Anonymous Monitoring Programme in England, conducted at least four times before and three times during and after the intervention. We will adapt Bayesian synthetic control methods (specifically the Causal Impact Method) to generate

### Strengths and limitations of this study

► Our control sites in the rest of Scotland and England were not randomised so there will be confounding and uncertainty in the intervention effect estimates.
► Hepatitis C virus treatment and prevention strategy in UK (and Europe) is evolving - motivated both by WHO 'elimination targets' and falling drug prices – which may contaminate our controls.
► However, our statistical models suggest that we should have sufficient power to detect an intervention effect and can model changes over time.
► We will develop dynamic transmission and economic models that can estimate cost-effectiveness including the prevention benefit of this intervention.
► We are conducting multiple nested qualitative studies and training and using peer researchers.

the cumulative impact of the intervention on chronic HCV prevalence and incidence. We will use a dynamic HCV transmission and economic model to evaluate the cost-effectiveness of the HCV TasP intervention, and to estimate the contribution of the scale-up in HCV treatment to observe changes in HCV prevalence. Through the qualitative data we will systematically explore key mechanisms of TasP real world implementation from provider and patient perspectives to develop a manual for scaling up HCV treatment in other settings. We will compare qualitative accounts of drug treatment and recovery with a 'virtual cohort' of PWID linking information on HCV treatment with Scottish Drug treatment databases to test whether DAA treatment improves drug treatment outcomes.

**Ethics and dissemination** Extending HCV community care pathways is covered by ethics (ERADICATE C, ISRCTN27564683, Super DOT C Trial clinicaltrials.gov: NCT02706223). Ethical approval for extra data collection from patients including health utilities and qualitative interviews has been granted (REC ref: 18/ES/0128) and ISCRCTN registration has been completed (ISRCTN72038467). Our findings will have direct National Health Service and patient relevance; informing prioritisation given to early HCV treatment for PWID. We will present findings to practitioners and policymakers, and support design of an evaluation of HCV TasP in England.

## INTRODUCTION AND BACKGROUND

Infection with hepatitis C virus (HCV) is a progressive disease that over 20 to 40 years can lead to liver cancer and premature death. HCV is the second largest contributor to liver disease in the UK and one of the few causes that is curable.[1] In the UK it is estimated that approximately 200 000 people are infected with HCV, over 85% of whom are people who inject or have injected drugs (PWID).[2–5] Chronic HCV prevalence and incidence among PWID remains high in UK at 20% to 50% and 5 to 15 per 100 person-years, respectively.[4 6–18] Prevention of HCV transmission among PWID is critical to long-term prevention of HCV related liver disease.[19]

We have reviewed the effectiveness of traditional primary prevention against HCV – opioid substitution treatment (OST) and needle and syringe programmes (NSPs).[12 20–22] Ongoing exposure to OST and high-coverage NSPs can reduce the risk of HCV transmission by 50% to 80%.[12 22] In Scotland HCV incidence among PWID decreased from approximately 14 to 6 per 100 person-years from 2008/2009 to 2011/2012 coinciding with the launch of the Scottish HCV strategy and action plan which incorporated scale-up of harm reduction interventions and HCV treatment.[10 23] We estimated that 60% of this decline could be attributed to the scale-up of OST and NSP during the action plan and that 1400 HCV infections were averted by 2015.[24] However, there was no appreciable reduction in overall anti-HCV prevalence over this short period, and there is some suggestion that incidence has increased recently to ~10 per 100 person-years (http://www.hps.scot.nhs.uk/resourcedocument.aspx?id=5863). HCV transmission models suggest that primary prevention through NSP and OST alone is insufficient to achieve substantial reductions (of the order of 40% or more within 10 years) in HCV prevalence among PWID in the UK.[25 26]

Prevention of hepatitis C disease and HCV transmission is now possible because highly effective, tolerable,

short-course interferon-free direct-acting antiviral therapies (DAAs) are available for all HCV genotypes with cure rates – defined as sustained virological response (SVR) - exceeding 90%.[27–29] We, and others, hypothesise that HCV treatment scale-up for PWID, and resulting HCV Treatment as Prevention (TasP) could enhance other primary interventions and reduce HCV incidence and chronic prevalence to negligible levels (ie, towards elimination as a major public health concern).[30–35] TasP refers to the concept whereby future transmission is reduced by treating affected individuals[36 37]: in HIV TasP antiretroviral treatment reduces transmission because individuals have undetectable infection[38] in HCV TasP people are cured so reducing opportunities for future transmission. WHO targets for HCV elimination, adopted by UK and other countries, aim to reduce HCV incidence by 80% and associated mortality by 65% by 2030.[39–44]

Clinical guidelines in Europe and USA changed from recommending prioritising HCV treatment to people with moderate-to-severe liver disease towards removing any restrictions and recommending that people at risk of transmission irrespective of fibrosis stage are offered treatment.[45–49] Cost-effectiveness models that incorporate the population prevention benefit suggest early treatment should be prioritised to PWID over other patient groups (unless chronic HCV prevalence and transmission is very high).[50] There is direct evidence that SVR following HCV treatment reduces liver disease progression and mortality risk,[51–53] but in two recent reviews we found no empirical evidence that HCV treatment scale-up has reduced chronic HCV prevalence and incidence in PWID populations.[36 37] In part this is because in most settings HCV treatment rates in PWID are too low and any changes generally too small to be detected, as we show in two studies of seven sites in UK[7] and an extension to 11 sites in Europe.[54] Until very recently in the UK, the annual number of HCV DAA treatments was restricted - as drug costs could be expensive (>£10 000 per patient). There is the opportunity now to test whether scaling up HCV treatment will reduce chronic HCV prevalence and transmission among PWID.[44]

In a pilot study ('Eradicate C') in Tayside we showed that we can increase HCV case-finding and engage and successfully treat PWID in the community.[55] Combining further studies on extending community HCV treatment pathways in Tayside and additional treatments provided by National Health Service (NHS) Tayside and Scottish Government we can establish an immediate natural experiment (with Tayside as the intervention site and other sites in Scotland and England as controls) to test and generate UK empirical evidence on the and potential impact and cost-effectiveness of HCV TasP in PWID. The UK is one of few countries worldwide to have an established nationwide surveillance system monitoring HCV infection among PWID.[9 12 17 22 56–60] This is undertaken through a series of cross-sectional voluntary anonymous surveys of PWID recruited at harm reduction services, referred to as the Unlinked Anonymous Monitoring

Programme (UAM) in England and Wales and the Needle Exchange Surveillance Initiative (NESI) in Scotland.[61 62] In addition, the UK has established sentinel laboratory surveillance of HCV testing and national monitoring of HCV treatment.[8 63–65] The data collected in both UAM and NESI will be used to assess out outcome.

Alongside a natural experiment in Tayside, we will collect information to assess the treatment facilitators and barriers. Historically it has proven very hard to engage PWID in HCV treatment.[66–69] Some barriers to engagement, such as poor efficacy or fear of interferon treatment side-effects, may be ameliorated by DAA therapy. However, other barriers such as mistrust of health services, stigma and competing priorities faced by PWID may persist. In addition, providers may be reticent to refer or provide HCV treatment to PWID due to concerns about adherence, reinfection and perceptions of treatment 'worth'.[70 71] It is expected that co-locating HCV treatment within existing services will reduce many system and provider level barriers to PWID accessing care.[66–68 72–77] However, this has not been tested in the context of community wide scale-up of interventions across multiple potential pathways. It is critical, therefore, that we understand how HCV TasP is embedded within the existing service landscape and incorporated into providers' professional roles.

Finally it has been hypothesised that successful HCV treatment in PWID may positively impact on understandings of self and identity and improve treatment of drug use disorders.[71 72 78–80] Accounts of 'transformative' outcomes extending beyond viral clearance alone include reference to reductions in drug and alcohol use, uptake of safer injecting practices, improved social relationships, enhanced sense of responsibility and self-worth. Hints of such collateral or indirect benefits are also found in quantitative studies reporting low re-infection rates and reductions in risky injecting behaviours among treated PWID.[81 82] We aim to test this hypothesis in our qualitative follow-up study and compare the findings to quantitative data generated from a virtual cohort.

## METHODS AND ANALYSIS
### Study design
Our intention is to create and conduct a mixed methods study, including qualitative studies and economic evaluation, of a natural experiment of HCV TasP among PWID. We also will develop methods for evaluating HCV TasP.

### Methods
#### Scaling-up HCV treatment
The **intervention** comprises the scale-up of HCV treatment in PWID which has started early in Tayside. By combining support from Scottish Government, National Health Board Tayside (NHS Tayside) and industry (MSD, Gilead, BMS) we can deliver rapid intensive scale-up of HCV treatments for PWID (comprising an extra 400 HCV treatments, a 3.5-fold increase from treatments for PWID prior to April 2017, see sample size below). We have developed multiple integrated community HCV care pathways, including novel care pathways in pharmacies, a low threshold NSP, drug treatment services and prisons (see figure 1). Our diagnostic pathways make extensive use of dried blood spot (DBS) testing for diagnosis of HCV antibody and chronic HCV with subsequent conventional laboratory testing in preparation for treatment (viral load, liver function and FIB-4 fibrosis score).[83–85]

### Study population
Our intervention is delivered and measured at the population level – which we have created by combining several individual studies and treatment pathways as shown in figure 1 (see ethics section below for the individual studies). We gained ethical approval from East of Scotland Research Ethics Service REC 1 (ref: 18/ES/0128) to ask patients for permission to be recruited

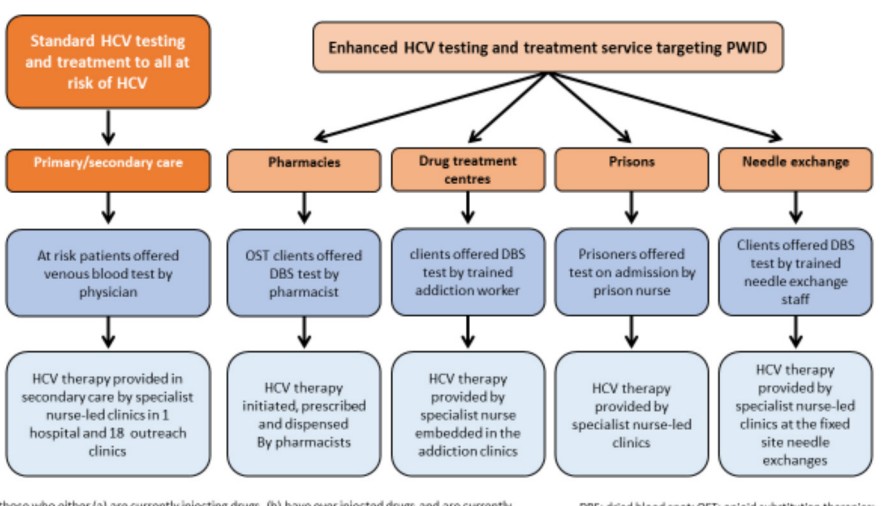

**Figure 1** Overview of HCV testing and treatment pathways for the PWID population in NHS Tayside. DBS, dried blood spot; HCV, hepatitis C virus; NHS,National Health Service; OST, opioid substitution treatment; PWID, people who inject drugs.

into the qualitative study (below) and extended clinical and behavioural drug history and data on health utilities (EQ5D-5L) at onset of treatment, during treatment and after the end of treatment.

Community HCV specialist nurses (3.5 full-time equivalent (FTE)) coordinate and deliver case-finding and treatment across the pathways in Tayside (figure 1).

The region of Tayside co-localises to NHS Tayside which is the provider of healthcare to a geographical area of 2903 sq mi (7519 km²) including the cities of Dundee and Perth and the counties of Angus and Perth & Kinross, situated in the east of Scotland with a population of 416 000. It is a mixture of urban and rural environments with some of the most affluent and most deprived areas in Scotland. It is therefore a representative microcosm of many areas in the UK.

### HCV treatment

Apart from expansion of community HCV care pathways, no new clinical procedures will be investigated and all PWID with chronic HCV will be offered oral DAA HCV treatment compliant with the Scottish clinical guidelines (https://www.hps.scot.nhs.uk/resourcedocument.aspx?id=6621).

As per local standard of care, participants will be offered appropriate harm reduction advice.

Standard care for patients is to test for SVR at 12 weeks after end of treatment with patients being recommended for annual follow-up if at risk of re-infection. Specialist nurses concentrate on building a good relationship with the participant to ensure that they do return for follow-up appointments. Health Protection Scotland collates national public health surveillance data on the number, characteristics and response of patients initiated onto HCV therapy, through Clinical Databases installed in 17 specialist HCV treatment centres, across Scotland.[41 86] A similar system also is available in England.

### HCV surveillance and intervention outcome (chronic HCV in PWID)

The **outcome** is chronic HCV prevalence (HCV viraemia as measured by HCV PCR) among PWID in the community (not just in the patients who undergo HCV treatment). Prevalence will be monitored using the NESI and UAM surveys, as detailed below. During 2017 to 2022, three waves of data collection for NESI (n=7500) and five to six for UAM (n=17 000 in England) will measure this outcome.

In our pre-intervention period from 2010/2011 to 2016 there have been four NESI surveys in Scotland (n=10 000 participants in total) and six UAM surveys in England (n=16 000 in total), which have involved the collection of DBS linked to questionnaire data. Participants are recruited at sentinel sites by a team of trained interviewers in Scotland (at over 100 NSP sites) and by agency staff in over 60 low-threshold drug agencies across England.[58 61] Participants complete a short questionnaire, with common questions across UAM and NESI, on demographics, injecting behaviour and service utilisation, and importantly (in relation to quantifying the intervention effect) both survey approaches have remained consistent over time.

The DBS samples collected in NESI and UAM have all been tested for HCV antibody, using the same methods (where sensitivity and specificity of the assay on DBS are close to 100%),[83 84] and illustrate that antibody prevalence (ever infection) has remained relatively stable among PWID during this time (figure 2). PCR positivity among antibody positive samples is used to determine chronic infection.

All NESI and UAM samples will be tested for HCV antibody and RNA PCR to assess the impact of HCV therapy scale-up – which is critical as trends in chronic infection and antibody status will diverge as more people are cured. In addition, we will undertake RNA PCR testing of all historical samples that were HCV antibody positive shown in figure 2 so that we can measure chronic

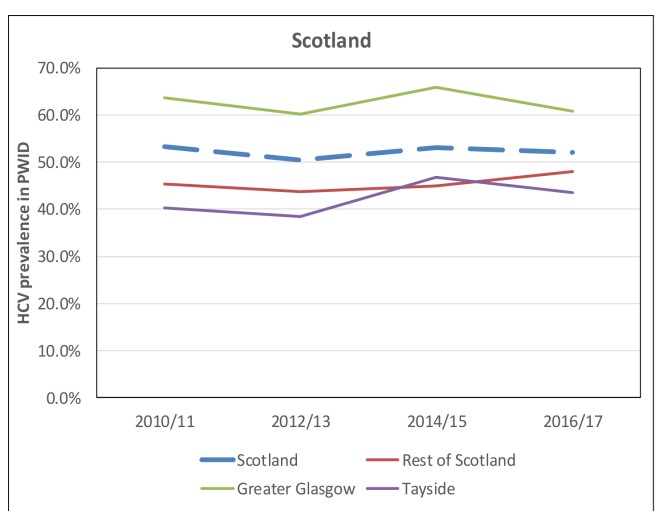
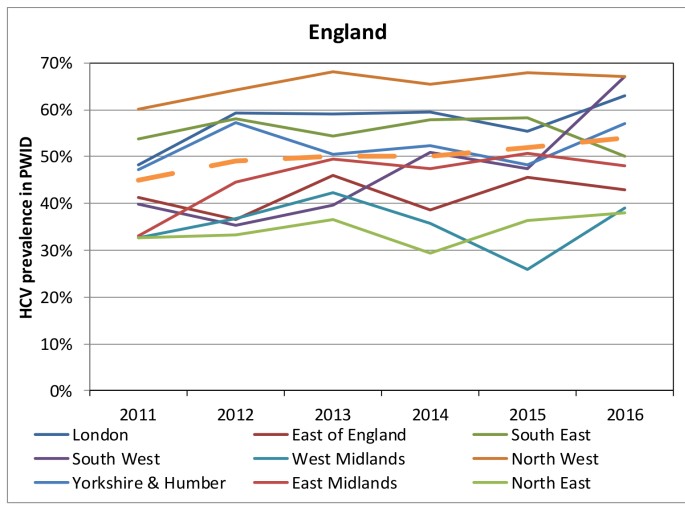

**Figure 2** Trends in HCV antibody prevalence among PWID in Scotland and England 2010/2011 to 2016. HCV, hepatitisC virus; PWID, people who inject drugs.

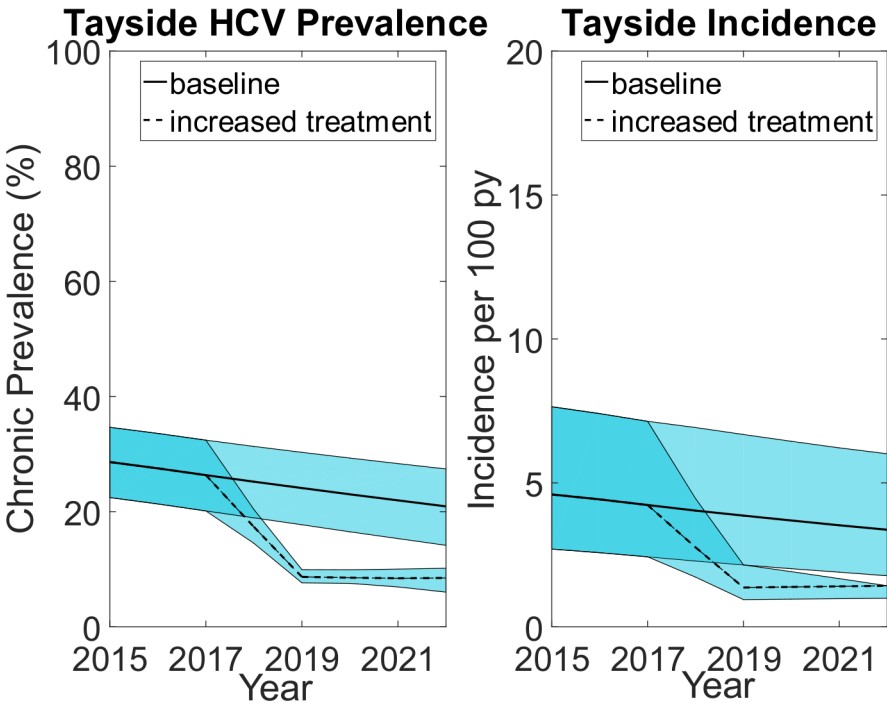

**Figure 3** Projected chronic HCV prevalence and incidence among PWID in Tayside with and without the intervention. Blue shaded area denotes the 95% credibility intervals of the model projections with and without the intervention. HCV, hepatitisC virus; PWID, people who inject drugs; py,person-years.

HCV prevalence among PWID pre-intervention, as well as post-intervention, for analysis (below)

Data on HCV PCR positivity among antibody negative samples identify recent infections and is used to estimate HCV incidence – which has fluctuated between 5 to 10 infections per 100 person-years across the UK during the last 5 years.[61] We will also estimate HCV incidence from our transmission dynamic models.[24 54]

### Sample size, power and estimating intervention effect

We updated estimates of the prevalence of PWID in Tayside[5] which suggest there are 2760 (95% Credible Interval, CrI 2360 to 3170) PWID either currently injecting and/or in OST. NESI data suggest that approximately 30% have chronic HCV and over 75% of PWID with chronic HCV have been diagnosed. Prior to 2017 approximately 66 PWID were treated annually. From April 2017 we plan to treat at least 500 PWID in Tayside over 2 years (as a result of expanded community care pathways shown in figure 1 and extra HCV treatments provided by NHS, Scottish Government and Industry funding). Adapting a transmission dynamic model that has been used in Tayside,[87] we hypothesise that within 2 years chronic HCV prevalence among PWID will reduce by approximately 62% from 26% (95% CrI 20 to 32) to at least 10% and chronic HCV incidence will fall by approximately 2/3s from 4.2 (95% CrI 2.4 to 7.1) per 100 person-years (p100py) to 1.4 (95%CrI 1.0 to 1.4) p100py (as shown in figure 3). Modelling also suggests that maintaining these reductions after 2019 will require less than 40 treatments per year.

We will adapt the Causal Impact synthetic control Model (CIM) as proposed by Brodersen and colleagues.[88 89]

We have performed simulation studies to test power and evaluate the utility of the CIM assuming information on chronic HCV prevalence among PWID (shown in figure 4). Provided trends in the chronic HCV prevalence in the pre-intervention period are relatively stable (which is the case) there will be sufficient power to detect the projected reduction in chronic prevalence. For example, in figure 4d we see that for a prevalence reduction of 40% by year 2 to 3 the credible intervals of the estimated cumulative effect (cumulative drop in prevalence) exclude zero, correctly identifying evidence of a successful intervention. Whereas a cumulative reduction of <20% is unlikely to be detected.

### Qualitative studies

#### Understanding the barriers and facilitators to scaling-up community-based HCV treatment

The qualitative study design has two distinct arms focusing on the intervention providers, and the intervention recipients.

#### Intervention providers

A purposive sample of 30 intervention providers, comprising nursing leads and key individuals from collaborating organisations will be approached directly by the lead hepatitis nurse. Seven focus groups will be convened according to professional role and locality:

► HCV healthcare specialists (nurses and physicians).
► Community pharmacists.

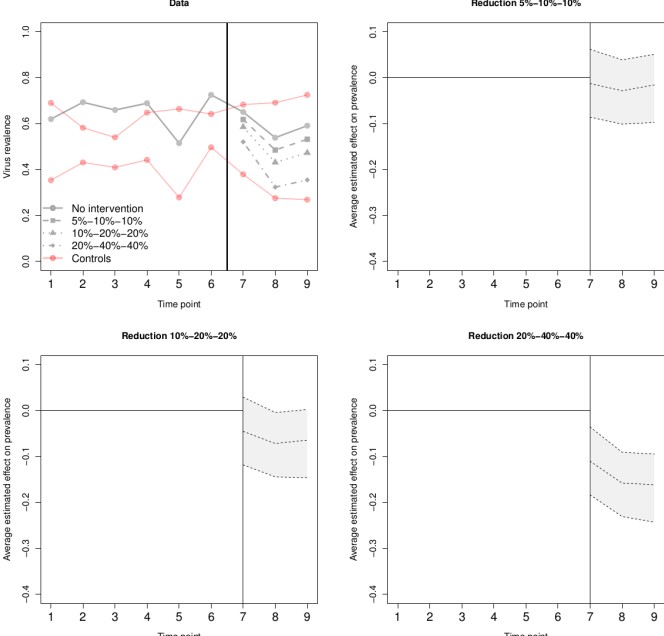

**Figure 4** Causal impact synthetic control method (CIM) simulation and estimated intervention effects and 95% credible intervals for a range of assumed effects. Footnote: Illustration of CIM. First subplot shows a single data set, where solid lines represent the simulated prevalence in the absence of the intervention, and the dashed lines represent the outcome of treated site in the post-intervention period under different intervention magnitude scenarios. For each one of the three scenarios, we calculate the estimated average intervention effect along with credible intervals. These are shown in subplots 2 to 4. We see that as the effect increases, the intervals tend to move away for zero. However, the intervention effect only becomes significant in scenario 3, where zero is not included in any of the post-intervention time points.

- ► Prison staff (both healthcare and security).
- ► 'Drug workers' (from OST and NSP services).

Each focus group will consist of a maximum of six individuals and ideally comprise multi-agency mixed groups. Individual interviews by telephone will be offered for those hesitant to join a group (estimate 10 interviews). Topic guides informed by previous work in this area[66 68 76 90] will facilitate group discussion.

### Intervention recipients – cross-sectional and longitudinal

The intervention recipient arm of the study will comprise both cross-sectional and longitudinal elements. A cross-sectional approach will be employed to recruit 6 to 10 participants who do not take up the offer of treatment. These individuals will be recruited through the treatment pathways or through our peer-researcher networks. The longitudinal element will follow a cohort of up to 40 individuals recruited following their course of HCV treatment. These individuals will be purposively sampled from the existing services in which HCV TasP has been embedded (ie, pharmacy, prison and drug service), and then followed-up at 1 year post-treatment (with 70% expected to be followed-up).[91] We aim to recruit women

as well as men, younger and older people; those treated previously and first time; those injecting and not injecting at treatment onset. Follow-up interviews will explore collateral effects of HCV TasP including outcomes pertaining to drug use and injecting practices (secondary outcome below).

Participants will be recruited by hepatitis nurses or other clinical staff in Tayside and the face-to-face semi-structured interview will be conducted by peer-researchers, trained and guided by experienced qualitative researchers. Dr Magdalena Harris explains the importance of the use of peer researchers within the context of EPIToPe: https://www.youtube.com/watch?v=9ZZo3fKOXlg.[92 93] The Scottish Drugs Forum (SDF) works with a group of Tayside peer-researchers with lived experience of injecting. Peer-researchers will receive study-orientated training and be provided with ongoing support to co-produce data and contribute to study outputs. A £20 shopping voucher will be offered to all interviewees except those in prison (Scottish prison service ethics did not permit thank you vouchers to prison participants).

### Qualitative data analysis

Interviews and focus-groups will be audio-recorded using encrypted digital voice recorders, transcribed verbatim and anonymised. *Nvivo* v.10 software will be used to code and manage qualitative data. First level analysis will be deductive, guided by the research questions, and peer researchers will be consulted for input and feedback during the analytical process.[94] A constant comparison method will be used to develop the thematic analysis and will reflect diverging and converging narratives, for example, across groups of intervention recipients at different time points in the treatment pathway, or between groups of intervention providers.[94] The findings will be contextualised in the relevant theoretical perspectives which may include the diffusion of preventive innovations (staff) or social norms and values that might underpin health behaviour (recipients).[95 96] We will assess TasP both from the providers' perspective and from patients' perspective including those who refuse treatment.

We will use the findings iteratively to update the HCV TasP logic model shown in figure 5. Our qualitative data will be used to generate a manual of an optimal intervention for other sites in UK. In previous examples, such as (https://www.youtube.com/channel/UCBV8smLmkO-QVT9D0OR-md1g/videos) we have used the Behaviour Change Wheel[96] as the framework to retrospectively analyse the success and failure of implementation within Tayside and then prospectively to formulate the optimal implementation intervention.

### Mixed method study on drug use outcomes: OST retention, drug overdose, recovery and social transformation

Health Protection Scotland (HPS) link data on diagnostic HCV tests in the four largest Scottish NHS boards (including Tayside)[8] and all persons undergoing HCV

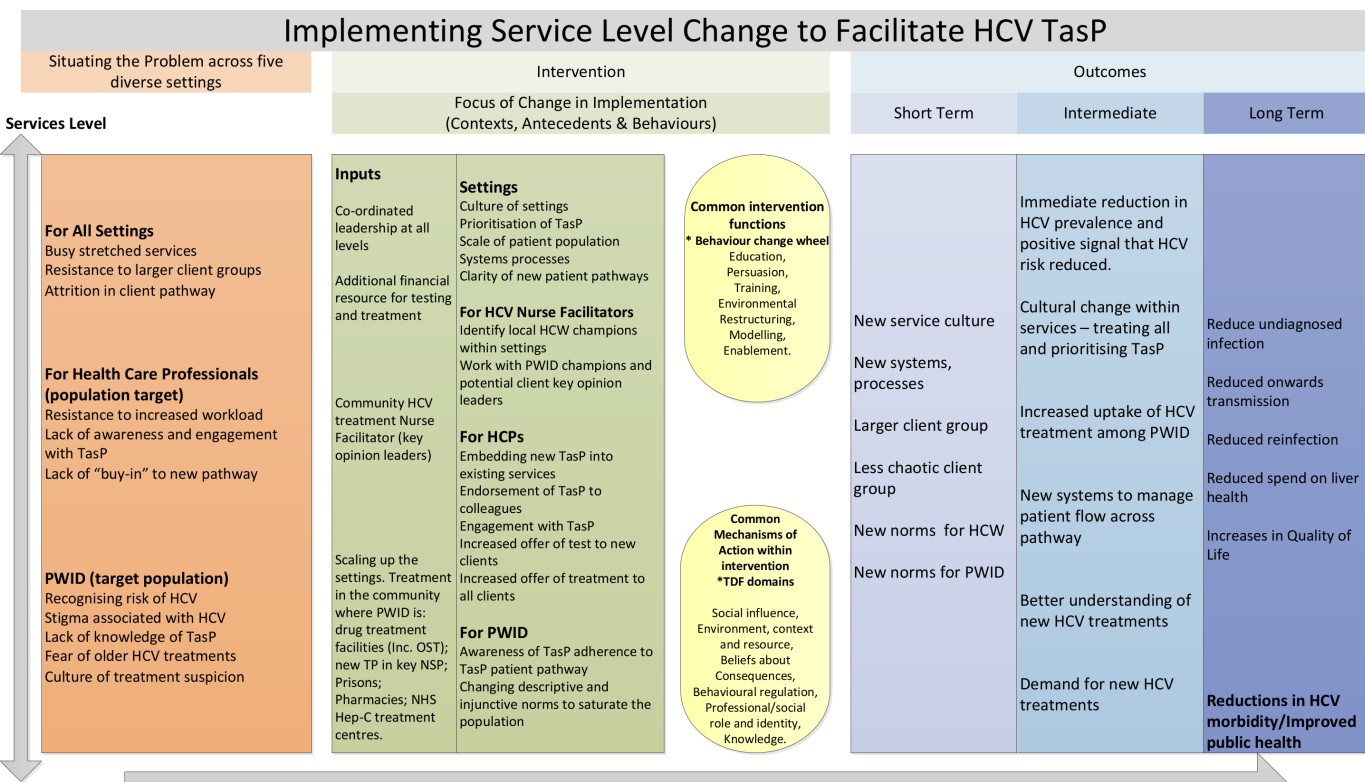

**Figure 5** Preliminary logic model HCV treatment as prevention (EPIToPe). HCPs, healthcare providers; HCV, hepatitis C virus; HCWs, healthcare workers; Hep-C, hepatitis C; NHS, National Health Service; NSP, needle and syringe programmes; TasP, Treatment as Prevention; PWID, people who inject drugs.

treatment in the Scottish HCV Clinical database[97] which are also linked with other databases (including deaths, hospitalisations and drug treatment)[8 42 98–100] and from 2018 Scotland's Prescribing Information System which holds data on OST and NHS prison health database (Prison Vision).[101–105] PWID attending drug services who were HCV diagnosed, compared with those who were not, are at increased risk of drug-related and other cause-specific morbidity/mortality.[106 107] Thus, we will create a virtual cohort of chronic HCV infected PWID (estimated to involve at least 600 individuals from Tayside and 3000 from elsewhere) and through linkage identify those who have been treated and attained SVR with those who have not. We will assess and compare the following outcomes: retention in drug treatment (determined through linkage to drug treatment and prescribing databases), drug-related and alcohol-related morbidity/mortality (through linkage to all hospital admission and mortality databases) and other markers of relapse (through linkage to prisons database).

### Economic and impact evaluation

Infectious disease models can test the extent to which observed changes in disease transmission can be attributed to specific interventions,[108–112] and assess cost-effectiveness of interventions that avert secondary infections, that is, have a population prevention benefit.[50 85 113–117] We will update and adapt a transmission model of HCV among PWID in Scotland and Tayside to model the impact of the HCV treatment intervention based on historical trends and new observations collected as part of this programme.[39 87] We will stratify the PWID population into current (injected in the previous year) and temporarily ceased (in OST and not injected in the previous year); as well as by duration of injecting (<3 years, 3 to 9 years, 10+years since onset), prevention intervention exposure (OST and/or high coverage NSP), and intervention settings for testing and treatment. We will use Approximate Bayesian Computation to calibrate the model to pre-intervention trends in chronic HCV prevalence and incidence among PWID in Tayside. The model will simulate the impact of observed rates of HCV treatment and cure rates for the intervention period, also incorporating any changes in the coverage of OST and NSP and injecting risk behaviours.

We will test consistency between the model impact projections and observed changes in HCV chronic prevalence and incidence from Tayside to disentangle the impact of HCV TasP from other interventions (OST/ NSP) or epidemiological changes, and predict the impact of the TasP on number of HCV infections averted. If they are not consistent then alternative evidence-based hypotheses will be tested for why the model projects a different impact and the best fitting models will then be used to project the impact of the intervention. This will

be assessed compared with two alternative counterfactuals where treatment rates are either at pre-scale-up levels in Tayside or at the average level achieved in other UK sites over the scale-up period. The impact of any changes in OST and NSP coverage will also be assessed to determine the contribution of those changes on observed effects. Impact will be assessed in terms of the relative decrease in prevalence and incidence, as well as the number and per cent of infections averted in the intervention model projections compared with each counterfactual over different time frames. These model projections can also be taken forward to evaluate the possible impact of the intervention over next 5 or 10 years.

We will evaluate the cost-effectiveness of the intervention (HCV treatment scale-up) compared with status quo (expected rate of HCV case-finding and treatment among PWID in the rest of the UK) from a healthcare provider (NHS) perspective, with the cost-effectiveness of the different settings where case-finding occurs also being assessed. The cost-effectiveness model will be based on the same dynamic impact model, adapted to include HCV disease progression stages and tracking of health outcomes among PWID after cessation of injecting.[50] The economic evaluation will incorporate both individual benefits of HCV treatment (on disease progression) as well as population benefits (on HCV transmission). We will calculate the total number of infections and deaths over a 50 year time horizon for the intervention and counterfactual scenario and estimate the costs and quality-adjusted life years (QALYs) based on the number of individuals in each disease stage per year in the model. We will discount all future costs and QALYs at 3.5% (The National Institute for Health and Care Excellence (NICE) guidelines https://www.nice.org.uk/process/pmg9/resources/guide-to-the-methods-of-technology-appraisal-2013-pdf-2007975843781). Probabilistic sensitivity analyses will be used to estimate the parametric uncertainty in the impact and cost projections. Cost-effectiveness results will be expressed in terms of incremental cost-effectiveness ratios and net monetary benefits estimated using NICE thresholds (£20 000 and £30 000 per QALY). We will plot cost-effectiveness acceptability curves to determine the probability of the intervention being cost-effective compared with different willingness-to-pay thresholds. Analyses of covariance methods will be used to summarise the proportion of the variability in the incremental costs and QALYs explained by uncertainty in different input parameters. Univariate sensitivity analyses will consider the effect of changes in important parameters such as time horizon, treatment cost and discount rate.

We focus on the incremental or additional resource costs associated with the intervention in Tayside. These costs, in part based on our earlier work for other studies, will include such things as the nurse time spent on intervention related activities (training other staff to offer HCV testing and treatment referral) as well as additional HCV testing and treatment costs, any additional OST costs due to HCV testing or treatment, and other staff time at the NSP, drug treatment centres and prisons involved with the intervention. Most of the incremental costs can be defined as variable (driven by extra nurse time and HCV testing/treatment costs). NHS HCV care costs and health utilities will be attached to each disease stage, based primarily on previous syntheses and models, which assume that PWID have a lower quality of life (QoL) than non-PWID of a similar age, gender and liver disease stage.[118–120] Additional data using the EQ-5D-5L tool during this study will generate new health utility data on the QoL among PWID before and after DAA treatment.

### Patient and public involvement

Patient and public involvement (PPI) was led by the Hepatitis C Trust and supported by qualitative research assessing barriers and facilitators to HCV treatment access (led by Dr Magdalena Harris). The SDF were also actively involved in the development of EPIToPe. The input from PPI groups has influenced the design of care pathways and has ensured that peer research is an essential element of the qualitative strand of EPIToPe.

A pilot National Institute for Health Research (NIHR) funded study in England (HEPCAT) responding to NICE Guidance on Hepatitis Case Finding was co-designed with Hepatitis C trust. It showed that Hepatitis C Facilitators and peer-support networks can increase the uptake of HCV case-finding and HCV treatment readiness in addiction services. This pilot study and our studies in Dundee/Tayside will influence how HCV treatment can be scaled up in England and our proposed evaluation HCV treatment as prevention.

Peer researchers will be trained to conduct the longitudinal study with PWID treated for HCV and will be involved and contribute to the analysis of the findings. Peer researchers and SDF will be members of the project management group and steering committee.

Dissemination events will be held in Dundee to discuss and present the findings from the qualitative studies with patient groups and services. These will be facilitated by SDF to support active contribution from our peer researchers. The study findings will be summarised and promoted through SDF website, social media platforms and through their sector-wide conferences in Scotland. Hepatitis Scotland, who are hosted within SDF, together with patient and public groups in England will take an active role in the wider national and international dissemination of the research, it's translation into patient meaningful materials and its integration into a national policy context. The research will also be promoted via Hepatitis C Trust and Public Health England.

### DISCUSSION
### Strengths and limitations of this study

Several limitations arise from the 'natural experiment' design as our intervention and controls were not randomised. In the UK and many other countries there is no longer sufficient equipoise in clinicians and policymakers

– given WHO and national strategies on HCV 'elimination' – to mount an randomised controlled trial of HCV TasP. As a result, there will be confounding and additional uncertainty in the measurement of the intervention effect. However, we consider that a natural experiment and use of synthetic control methods to be a more robust design than simple before and after studies. Our preliminary simulation work also suggests that we should have sufficient power to detect the large intervention effect that is planned.

We know also that HCV treatment and prevention strategy in UK (and Europe) is evolving – motivated both by WHO 'elimination targets' and falling drug prices – and our control sites in Scotland and England may increase HCV treatment rates earlier than expected. This will complicate the analyses a little and potentially dilute the intervention effect. We are confident that we can adapt the synthetic control methods to take account of changes over time – and that because Tayside has started so early in scaling up HCV treatment that we will have time to detect a difference in the outcome.

The lack of randomised controls means that we have to generate the counterfactual of 'no HCV treatment scale-up' through our HCV transmission model so that we can subsequently estimate cost-effectiveness of the intervention in Tayside. This is not ideal but has become standard practice in economic models of novel HCV treatment interventions – and we are involved with the modelling of HCV treatment pathways through homeless centres, prison, Accident and Emergency (A&E), pharmacies, specialist drug clinics and NSPs (P Vickerman personal communication and for example[55 85 121]). We know also, however, that the benefit in terms of additional Quality of Life Years and averted HCV infections accrues and occurs over a prolonged period.[50] It is more critical for any economic evaluation of HCV interventions in PWID that a dynamic model is used so that the prevention benefit (in terms of HCV infections averted) is correctly accounted for.

We are using peer researchers in the qualitative arm of patients' perspectives on the intervention and on the impact of HCV treatment on addiction outcomes. This is novel but adds additional challenges to obtaining NHS passports and ensuring data quality across the interviews and interviewees. We are also intending to support peers in analysis and interpretation of the findings which we believe has not been done before. We have trained the interviewers and will be monitoring their performance of the interviewers to ensure consistent study quality – and will replace peers with our qualitative researcher if required.

## Future study: natural experiment of TasP in England

In England HCV treatment is delivered through 22 operational delivery networks (ODNs). NHS England's HCV strategy (2016 to 2019) prioritised 10 000 patients per year in line with the declared priorities of the network which could (and in many cases did) include people who use drugs at risk of transmission.[44] In October 2018 it is anticipated that a new procurement deal will substantially

increase the number of patients who can access DAAs and this will enable 'trace and treat' options to be introduced. We will use the first part of EPIToPe including the manual generated by the qualitative study, enhancements to historical and ongoing surveillance of chronic HCV in PWID, infectious disease models and methodological developments of causal impact model, to co-design with ODN leads a natural experiment of HCV TasP in England.

**Author affiliations**
[1]Population Health Sciences, Bristol Medical School, Bristol, Bristol, UK
[2]Hepatology & Gastroenterology, Clinical & Molecular Medicine, School of Medicine, University of Dundee, Dundee, UK
[3]Glasgow Caledonian University, Glasgow, UK
[4]MRC Biostatistics Unit, School of Clinical Medicine, University of Cambridge, Cambridge, UK
[5]Blizard Institute, Queen Mary University of London, London, UK
[6]Barts Health NHS Trust, London, UK
[7]Dundee Epidemiology and Biostatistics Unit, University of Dundee, Dundee, UK
[8]Tayside Health Board, Dundee, UK
[9]Health Protection Scotland, Glasgow, UK
[10]National Infection Service, Public Health England, London, UK
[11]Scottish Drugs Forum, Edinburgh, UK
[12]Division of Infectious Diseases and Global Public Health, University of California San Diego, San Diego, UK
[13]Tayside Clinical Trials Unit, Tayside Medical Science Centre, University of Dundee, Dundee, UK
[14]ISD Scotland, Edinburgh, UK
[15]West Of Scotland Specialist Virology Centre, NHS Greater Glasgow & Clyde Board, Glasgow, UK
[16]Scottish Drug Forum, Edinburgh, UK
[17]London School of Hygiene and Tropical Medicine, London, UK
[18]Liverpool John Moores University, Liverpool, UK
[19]Directorate of Public Health, NHS Tayside, Dundee, UK
[20]Edinburgh Napier University, Edinburgh, UK

**Acknowledgements** We would like to thank all those involved in PPI from both SDF and the Hepatitis C Trust.

**Contributors** All authors contributed to editing of the manuscript. MH and SJH are co-PIs of EPIToPe and prepared first draft of the manuscript. JD leads intervention scale-up in Tayside in collaboration with Tayside CTU LJZB, PD, SKI, AR and AE. LE leads qualitative component of EPIToPe in collaboration with Scottish Drug Forum DL, EH and AM and support from MH, GV and DW on qualitative research and training of peer support workers, and PF on behavioural science. DDA leads synthetic control estimation and multiple parameter evidence synthesis in collaboration with PS, RH, AP, and NM. PV leads dynamic impact and economic modelling in collaboration with NM, ZW, HF with health economics led by W Hollingworth (WH), with GM, as part of Bristol Randomised Trial Collaboration (BRTC) with advice on trial design from JH, CM and AL. GF is leading design of evaluation in England based on EPIToPe with support from BRTC and KD. SH is leading on outcome measurement in Scotland with Health Protection Scotland DG, AMA, and in collaboration with LG from ISD, RNG, HI, NEP and AY. SM and SI are leading on outcome measurement in England with RG, HH, EH, VH, SM, MR, RS, SK. JM is the Programme Manager.

**Funding** 'This study is funded by the National Institute for Health Research (NIHR) Programme Grants for Applied Research programme (Grant Reference Number RP-PG-0616-20008). The views expressed are those of the author(s) and not necessarily those of the NIHR or the Department of Health and Social Care '. In addition, we acknowledge support from NIHR Health Protection Research Unit in Evaluation, and the Bristol Randomised Trials Collaboration (BRTC), a UKCRC Registered Clinical Trials Unit in receipt of NIHR Clinical Trials Units (CTU) support funding. The views expressed are those of the author and not necessarily those of the NHS, the NIHR or the Department of Health and Social Care. NM is supported by the National Institute for Drug Abuse [grant number R01 DA037773] and the University of California San Diego Center for AIDS Research (CFAR), a National Institute of Health (NIH) funded program [grant number P30 AI036214].

**Patient consent for publication** Not required.

**Provenance and peer review** Not commissioned; externally peer reviewed.

**Data availability statement** All data relevant to the study are included in the article or uploaded as supplementary information.

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
