## [Reviewer comments · BMJ Open]

ARTICLE DETAILS

TITLE (PROVISIONAL)	Evaluating the Population Impact of Hepatitis C Direct Acting Antiviral Treatment as Prevention for People Who Inject Drugs (EPIToPe) – a natural experiment (Protocol)
AUTHORS	Hickman, M; Dillon, John; Elliott, Lawrie; De Angelis, Daniela; Vickerman, Peter; Foster, Graham; Donnan, Peter; Eriksen, Ann; Flowers, Paul; Goldberg, David; Hollingworth, William; Ijaz, Samreen; Liddell, David; Mandal, Sema; Martin, Natasha; Beer, Lewis; Drysdale, Kate; Fraser, Hannah; Glass, Rachel; Graham, Lesley; Gunson, Rory; Hamilton, Emma; Harris, Helen; Harris, Magdalena; Harris, Ross; Heinsbroek, Ellen; Hope, Vivian; Horwood, Jeremy; Inglis, Sarah; Innes, Hamish; Lane, Athene; Meadows, Jade; McAuley, Andrew; Metcalfe, Chris; Migchelsen, Stephanie; Murray, Alex; Myring, Gareth; Palmateer, Norah E.; Presanis, Anne; Radley, Andrew; Ramsay, Mary; Samartsidis, Pantelis; Simmons, Ruth; Sinka, Katy; Vojt, Gabriele; Ward, Zoe; Whiteley, David; Yeung, Alan; Hutchinson, Sharon

VERSION 1 – REVIEW

REVIEWER	Asher Schranz, MD University of North Carolina, Chapel Hill, North Carolina, USA
REVIEW RETURNED	13-Mar-2019

GENERAL COMMENTS	This submission is the protocol for a large, multifaceted real-world study of the treatment of Hepatitis C among people who inject drugs in Tayside, Scotland. It aims to generate pragmatic evidence on HCV treatment as prevention among PWID. Notable strengths of the protocol include:  • A large, real-world effectiveness study. • Existing familiarity of the researchers with Tayside through prior work, engaging relevant stakeholders across numerous sectors (eg prison, OST, NSP staff, etc) and prior modeling studies informing specific estimates of the expected effect magnitude. • Outcome is clearly defined. • Strong existing surveys providing baseline and ongoing data (NESI and UAM) • Use of peer researchers • Inclusion of a comprehensive logic model • Discussion of how work in this study will inform a subsequent natural experiment study in England. • Great figures that translate complex analysis into clear messages.
--

	Some specific comments:  - Page 5, Line 7: HCV is referred to as the “second most important cause of liver disease in the UK.” The use of the word “important” can seem vague and subjective, and it may be better to clarify whether this importance is due to prevalence, incidence, transmissability, etc. - Page 5, Line 37: The authors may consider adding 1-2 sentences explaining the concept of “treatment as prevention” as some readers may not be familiar with it. - Page 7, Line 39: The authors reference ethical approval to recruit patients into aspects of the study. Is this akin to an institutional review board that granted this approval? Can you state which body gave the ethical approval? - Page 8, Line 40: I found it surprising that the authors estimate that only 30% of PWID in Tayside have chronic HCV. This number seems somewhat low. Can this estimate be further explained? Can you add a sentence summarizing prior work that demonstrated this estimate? Is this number lower than– or consistent with–other comparable regions in the UK? - Page 10, Line 44: It is unclear why this Youtube video link is included here. - Page 11, Line 11: There is a large amount of diverse methodology in this protocol. Therefore, any given reader will likely be unfamiliar with at least a component – whether the modeling and analytic components, the qualitative framework, etc. Here, can you give some additional background on the Behavior Change Wheel and Theoretical Domains Framework? What are the major elements to these frameworks and are there a couple examples of how they may be applied to assess implementation? - Generally, is it possible to further flesh out the limitations of this study? They are referenced at the end of the abstract but otherwise are not discussed in-depth. - Can you add some text about Tayside generally, in terms of sociodemographics, employment, economy, infrastructure, etc? Can a comment be made about the expected generalizability of the experience Tayside to other settings in the UK or elsewhere?
--	---

REVIEWER	Benjamin Linas Boston University, USA
REVIEW RETURNED	19-Mar-2019

GENERAL COMMENTS	This manuscript describes the protocol for a new and exciting study in the U.K. that seeks to expand HCV treatment among persons who inject drugs (PWID) and decrease the prevalence of HCV among PWID. The study is very exciting and innovative and will play a major role in developing the evidence base about HCV cure for prevention among drug users. The manuscript, however, is too long and I found it very difficult to read, despite the fact that this is my field of interest and appears to be the dream study. If I cannot read this paper, then no one can.
--

	The protocol is to expand HCV treatment among PWID in one community, and to then use HCV surveillance data to compare the prevalence of HCV over time in that community to control communities that are not exposed to the intervention. In addition, the team will conduct qualitative interviews with both PWID and providers to learn more about barriers and facilitators to care, and they will use simulation modeling to project long-term outcomes and cost-effectiveness. This is basically all I can say about this study after reading the manuscript. Major details that are currently missing:  1. The paper provides almost no detail of the intervention itself, other than to say that they will work in places where PWID live or visit and treat more of them. How? What is the recruitment plan? Is the plan for individual therapy? Group treatment sessions? Will there be any effort to address substance use disorder at the same time as HCV? Will you offer medications for opioid use disorder with HCV treatment? How will you track and monitor treatment? What is the plan for securing end of treatment and SVR blood work (which is notoriously difficult to attain even with surveillance systems in place). 2. Primary outcome – how will you measure it? It is clear that you plan to use surveillance data which contain HCV Ab, but that is not the right test for prevalence. If you plan to use HCV Ab, you will run in to the problem that the apparent incidence could paradoxically RISE when you intervene, because people begin to seek testing and treatment and chronic HCV cases appear as new cases in the surveillance data. Further, if Ab is the only outcome measure, then you will not see a positive trial, because patients with SVR maintain POS HCV Ab for life. I am nearly certain that the team understands these points and does have a plan for using RNA, because there is a brief section on RNA results, but I do not understand how the study plans to use the combination of AB and RNA as an outcome and where all of those data come from. [As a scientific point (as opposed to protocol comment) – if you have Ab and RNA on all PWID, that is very valuable and could provide opportunity to measure true incidence, as well as the instantaneous prevalence of acute HCV. Those would be very valuable outcome.] Some ideas for where the authors could remove text to make the manuscript more accessible and also focus more sharply on the study protocol.  1. Pg 9 (using the page numbers inserted by the editorial system in the top left corner, not the numbers provided in the submission itself) Line 45- Pg 8 line 18 – this is the outcomes section, but there is a lot of text about preliminary studies using DBS and a lot of detail about the UK HCV surveillance system. I appreciate that you plan to use the surveillance data as your primary outcome measure and it is therefore important. That being said, this can be much more succinct. The primary outcome is HCV prevalence. You will measure prevalence among all PWID in the community using national surveillance data. 2. Pg 11 line 48 Pg 12 line 4 – this is all introduction text and it does not belong in the methods section. Much of it is redundant and could be cut. In this section of the paper, you should not justify outcomes or explain the background of your methods. All of that
--	---

	can go into the introduction, which frees you to get to the point in the methods. 3. Page 13 line 24 – 34 – again, this is background and does not belong in the methods 4. The simulation modeling plan – This is a major strength of the study, but it is too much in the paper. I think that you should be able to summarize the modeling plan and the outcomes in a paragraph or so. This is not the grant proposal, nor is it the methods section of the modeling manuscripts. You could likely generate a protocol paper just about the modeling, but that is not appropriate in this paper. Keep it short. I cannot give the manuscript a full edit in this manner, but the above examples provide a sense of the type of text that can be condensed, removed, and reorganized for a tighter paper and an easier read. In addition, because this is a mixed methods study with a variety of outcomes and data sources, I think that the paper could use a table that summarized outcomes as follows: Outcome Measure Data Source Hypothesis Prevalence of HCV In summary, I look forward very much to seeing the results of this study and I suspect that I will incorporate their findings into my own work. This protocol paper, however, needs a good edit to remove a lot of words, focus more sharply on the protocol and less on background, and make the outcomes more clear.
--	--

VERSION 1 – AUTHOR RESPONSE

Section	Comment	Response
Title	Please make it clear in the title that this is a protocol	Added to Title
Abstract	Please only include one abstract in your manuscript, containing the following sections: Introduction, Methods, Ethics & Dissemination. Please try and keep this as concise but informative as possible.	Now only one abstract with the recommended sub-headings
Intro and abstract	HCV is referred to as the “second most important cause of liver disease in the UK.” The use of the word “important” can seem vague and subjective, and it may be better to clarify whether this importance is due to prevalence, incidence, transmissability, etc.	We have amended text to second largest contributor.

Abstract and Discussion	Generally, is it possible to further flesh out the limitations of this study? They are referenced at the end of the abstract but otherwise are not discussed in-depth.	We have added discussion section.
Intro	The authors may consider adding 1-2 sentences explaining the concept of “treatment as prevention” as some readers may not be familiar with it.	Have added a sentence.
Methods	The paper provides almost no detail of the intervention itself, other than to say that they will work in places where PWID live or visit and treat more of them. How? What is the recruitment plan? Is the plan for individual therapy? Group treatment sessions? Will there be any effort to address substance use disorder at the same time as HCV? Will you offer medications for opioid use disorder with HCV treatment? How will you track and monitor treatment? What is the plan for securing end of treatment and SVR blood work (which is notoriously difficult to attain even with surveillance systems in place).	We apologise for the lack of clarity. We have created sub sections: study design, study population, HCV treatment and HCV surveillance. Our intervention (and outcome) - HCV treatment scale-up in the population (and chronic HCV in PWID) – is delivered and measured at the population level. This is why we highlight the UAM/NESI as they are critical to evaluating the natural experiment. The points raised by the reviewer refer more to individual studies of care pathways with SVR as the outcome. This is not our intention – and we are not altering the way HCV treatment or adjunct OAT is delivered under current guidance and recommendations. There is a national HCV treatment database that we can use to track treatments delivered – which we have added.

		If our revisions are not clear we will be glad to expand.
Methods	The authors reference ethical approval to recruit patients into aspects of the study. Is this akin to an institutional review board that granted this approval? Can you state which body gave the ethical approval?	Review board information added
Methods	- Can you add some text about Tayside generally, in terms of sociodemographics, employment, economy, infrastructure, etc? Can a comment be made about the expected generalizability of the experience Tayside to other settings in the UK or elsewhere?	Many thanks – we have added a paragraph.

HCV surveillance /Outcome	Primary outcome – how will you measure it? It is clear that you plan to use surveillance data which contain HCV Ab, but that is not the right test for prevalence. If you plan to use HCV Ab, you will run in to the problem that the apparent incidence could paradoxically RISE when you intervene, because people begin to seek testing and treatment and chronic HCV cases appear as new cases in the surveillance data. Further, if Ab is the only outcome measure, then you will not see a positive trial, because patients with SVR maintain POS HCV Ab for life. I am nearly certain that the team understands these points and does have a plan for using RNA, because there is a brief section on RNA results, but I do not understand how the study plans to use the combination of AB and RNA as an outcome and where all of those data come from.	Apologies for not being clear. The reviewer is absolutely right we are measuring Chronic HCV by RNA and have amended the text.
Outcome	this is the outcomes section, but there is a lot of text about preliminary studies using DBS and a lot of detail about the UK HCV surveillance system. I appreciate that you plan to use the surveillance data as your primary outcome measure and it is therefore important. That being said, this can be much more succinct. The primary outcome is HCV prevalence. You will measure prevalence among all PWID in the community using national surveillance data.	We have cut some of the text and moved some detail to intro/ background
Outcome	In addition, because this is a mixed methods study with a variety of outcomes and data sources, I think that the paper could use a table that summarized outcomes as follows: Outcome Measure Data Source	We were not sure whether a table would help. Instead we have added more subsections to the methods.

	Hypothesis Prevalence of HCV	
Sample size, Power, and Estimating Intervention Effect	I found it surprising that the authors estimate that only 30% of PWID in Tayside have chronic HCV. This number seems somewhat low. Can this estimate be further explained? Can you add a sentence summarizing prior work that demonstrated this estimate? Is this number lower than– or consistent with– other comparable regions in the UK?	These are observed data from NESI – which we now explain.
Sample size, Power, and Estimating Intervention Effect	The simulation modeling plan – This is a major strength of the study, but it is too much in the paper. I think that you should be able to summarize the modeling plan and the outcomes in a paragraph or so. This is not the grant proposal, nor is it the methods section of the modeling manuscripts. You could likely generate a protocol paper just about the modeling,	We have reduced the text.

	but that is not appropriate in this paper. Keep it short.	
Qual study	this is background and does not belong in the methods	Moved to intro/ background
Qual studies	this is all introduction text and it does not belong in the methods section. Much of it is redundant and could be cut. In this section of the paper, you should not justify outcomes or explain the background of your methods. All of that can go into the introduction, which frees you to get to the point in the methods.	Moved to intro/background and linked
Qual studies	It is unclear why this Youtube video link is included here.	Have added a sentence to explain.
Qual studies	There is a large amount of diverse methodology in this protocol. Therefore, any given reader will likely be unfamiliar with at least a component – whether the modeling and analytic components, the qualitative framework, etc. Here, can you give some additional background on the Behavior Change Wheel and Theoretical Domains Framework? What are the major elements to these frameworks and are there a couple examples of how they may be applied to assess implementation?	We have added another Youtube to show an example of a manual using similar process. We have referenced the Behaviour Change Wheel (which is reasonably well known) and hope that is sufficient (with the link to example). Of course, we will be glad to provide further information if required.
Mixed methods study	again, this is background and does not belong in the methods	Moved to intro

VERSION 2 – REVIEW

REVIEWER	Asher Schranz University of North Carolina, Chapel Hill, NC, USA
REVIEW RETURNED	15-May-2019

GENERAL COMMENTS	The authors' edits are appreciated and the article is streamlined. This remains a fascinating, complex and multifaceted protocol for a study that will likely have a broad impact. While I appreciate the editing of some of the technical text around the synthetic control model, the concepts and statistics in this section remain too advanced for me. Regarding the discussion of the synthetic control, the causal impact method and Figure 4, I would defer to a statistician for review, or another reviewer who
---

	feels she or he has competency to evaluate the appropriateness of this evaluation.
--	--